

## 2  COMPREHENSIVE LANDSLIDE SUSCEPTIBILITY MAP OF CENTRAL ASIA

Ascanio Rosi[a,b], William Frodella[b,c,*], Nicola Nocentini[b,c], Francesco Caleca[b,c], Hans Balder Havenith[d], Alexander
Strom[e,f], Veronica Tofani [b,c]
[a] Department of Geosciences, University of Padua, Via G. Gradenigo 6, 35131, Padua, Italy
[b] UNESCO Chair on the Prevention and Sustainable Management of Geo-Hydrological Hazards, University of
Florence, Largo Fermi 2, 50125 Florence, Italy
[c] University of Florence, Department of Earth Sciences, via G. la Pira 4, 50121 Florence, Italy.
[d] Department of Geology, University of Liege, 4000 Liege, Belgium
[e] Geodynamics Research Center LLC, Moscow, 125008, Russian Federation
[f] Geodynamics Research Center - branch of JSC "Hydroproject Institute", Moscow, 125993, Russian Federation
*       Correspondence: william.frodella@unifi.it; +39 055 2755979
**Abstract**
Central Asia is an area characterized by complex tectonics and active deformation; the related seismic activity
controls the earthquake hazard level that, due to the occurrence of secondary and tertiary effects, has also direct
implications on the hazard related to mass movements as landslides, which are responsible for an extensive number
of casualties every year. Climatically, this region is characterized by strong rainfall gradient contrasts, due to the
diversity of climate and vegetation zones. The region is drained by large, partly snow- and glacier-fed rivers, that
cross or terminate in arid forelands; therefore, it is affected also by a significant river flood hazard, mainly in spring
and summer seasons. The challenge posed by the combination of different hazards can only be tackled considering
a multi-hazard approach harmonized among the different countries, in agreement with the requirements of the
Sendai Framework for Disaster Risk Reduction. This work was carried out within the framework of the SFRARR
Project ("*Strengthening Financial Resilience and Accelerating Risk Reduction in Central Asia*") as a part of a
multi-hazard approach, and is focused on the first landslide susceptibility analysis at a regional scale for Central
Asia. To this aim the most detailed landslide inventories, covering both national and transboundary territories were
implemented in a Random Forest model, together with several independent variables. The proposed approach
represents an innovation in terms of resolution (from 30 to 70 m) and extension of the analysed area with respect
to previous regional landslide susceptibility and hazard zonation models applied in Central Asia. The final aim
was to provide a useful tool for land use-planning and risk reduction strategies to landslide scientists, practitioners
and administrators.
**1. Introduction**
During the two decades spanning between 1988 and 2007, according to observed estimates, out of 177 reported
disasters in Central Asia 13% were landslides, causing 700 deaths (Table 1), while in the same period economic
losses have been as high as $150 million, including damage to infrastructures, settlings and agricultural/pasture





lands, as well as displacement of the population (GFDRR, 2009). More recent modelled estimates show that in the
Central Asia states an annual average of 3 million persons are affected by earthquakes and floods, with an estimated
annual average GDP of 9 billion USD (GFDRR, 2016).
**Table 1: Observed landslide hazard statistics (1988-2007)**. Source: Risk assessment for Central Asia and
Caucasus (UN ISDR, 2009).

| Country | No. disasters/year | Total no. of deaths | Deaths/year | Relative vulnerability (deaths/year/million) |
|---------|-------------------|---------------------|-------------|----------------------------------------------|
| Kazakhstan | 0.05 | 48 | 2.40 | 0.16 |
| Kyrgyz Republic | 0.30 | 238 | 11.90 | 2.27 |
| Tajikistan | 0.50 | 339 | 16.95 | 2.51 |
| Turkmenistan | n.a. | n.a. | n.a. | n.a. |
| Uzbekistan | 0.15 | 75 | 3.75 | 0.14 |


Due to their large size and impact, most of the occurring landslides have profound transboundary implications.
Tajikistan and Kyrgyz Republic are the countries most impacted by landslides: in Tajikistan around 50000
landslide were mapped, 1,200 of which threaten settlements or facilities (Thurman, 2011), while Kyrgyz Republic
has been affected by 5,000 landslides, of which 3,500 at various levels of activity are located in the southern
portion of the country (the Fergana Valley area) (Pusch, 2004; Li et al., 2021). Only in Kyrgyz Republic, up to
2017, 784 landslides and 1658 mudflows (also including loess flows) and flash floods caused 352 victims
(Kalmetieva et al., 2009; Havenith et al., 2015a; 2017). Almaty province in Kazakhstan, Tashkent, Samarkand,
Surkhandarya, Kashkadarya Provinces of Uzbekistan, and Ahal Province of Turkmenistan are also exposed to
landslides (World Bank, 2006). Given the increased anthropogenic pressures and the impact of climate change,
since the early '90s several projects have tried to improve the knowledge on landslide hazard (Thurman, 2011),
by providing landslide losses estimations, location, type, triggering/reactivation dates, inventories and hazard/risk
maps, as well as platforms to retrieve open disaster risk data and overviews on landslide risk reduction strategies.
Amongst the regional studies on landslide hazard, providing descriptions, statistics, and inventory maps, it is worth
mentioning:
• Disaster Risk Management and Climate Change Adaptation in Europe and Central Asia, developed by the
World Bank - Global Facility for Disaster Reduction and Recovery (Pollner et al., 2010).
• Disaster Risk Reduction, 20 Examples of Good Practice from Central Asia, developed by the European Union,
International Strategy for Disaster Reduction ISDR (European Commission Humanitarian Aid, Civil
Protection, 2006).
• Science for Peace Project (983289) 'Prevention of landslide dam disasters in the Tien Shan, LADATSHA'.
2009–2012, NATO Emerging Security Challenges Division.



- PROGRESS (Potsdam Research Cluster for Georisk Analysis, Environmental Change and Sustainability). German Federal Ministry of Research and Technology (BMBF).
- Tian Shan-Pamir Monitoring Program (TIPTIMON). German Federal Ministry of Education and Research (BMBF).
- M126 IPL Project (funded by the International Consortium on Landslides): M2002111 Detailed study of the internal structure of large rockslide dams in the Tien Shan; M2004126 Compilation of landslide/rockslide inventory of the Tien Shan Mountain System.

Besides the creation of landslide inventories, a common approach to assess landslide hazard is the development of landslide susceptibility maps (LSMs), which depict the relative probability of occurrence of a given type of landslide in a given area, without considering the probability of occurrence in time (Brabb, 1984). In other words, LSMs identify those areas where landslides can occur, based on their geological, morphological, and climatic characteristics. These maps have been extensively used as useful tools for land planning (Cascini 2008; Frattini et al., 2010) and hazard assessment (Corominas et al., 2003). More recently, they have been successfully integrated also in quantitative risk assessment (Chen et al., 2016), and early warning systems (Segoni et al., 2018: Tiranti et al., 2019). LSMs have been produced by applying a wide range of mathematical techniques, from the most traditional statistic approaches like frequency ratio (Yilmaz, 2009), discriminant analysis (Carrara, 1983; Trigila et al., 2013) and logistic regression (Lee, 2005; Duman et al., 2006; Manzo et al., 2013), to more recent and more advanced techniques, like artificial neural network (Tien Bui et al., 2016; Ermini et al., 2005), machine learning (Catani et al., 2013) and multi criteria decision analysis (Akgun, 2012). Statistical-probabilistic models for landslide susceptibility can overcome the data gaps and allow to analyse very wide areas (from basin to national scales), by adopting a homogeneous methodology and a harmonized dataset (including global and local data sources). However, landslide hazard assessment is a complex process since it needs accurate knowledge of the topic and appropriate input data (historical inventories, and regional inventories that consist of large prehistoric events mainly). In this work the landslide susceptibility analysis was carried out by means of the "Random Forest" machine learning algorithm, which is credited as one of the most advanced and reliable techniques in this field (Catani et al., 2013, Goetz et al 2015). This work represents the first landslide susceptibility analysis at a regional scale for Central Asia, and was carried out in the framework of the SFRARR Project ("Strengthening Financial Resilience and Accelerating Risk Reduction in Central Asia") as a part of a multi-hazard approach (Bazzurro et al., in prep). The main challenge of this work was the creation of a unique LSM of the whole Central Asia, that involved the use of a wide range of variables, to account the features of each country, a high volume of input data, and the development of new approaches to analyse these data and to take into accounts possible discrepancies and non-homogeneities. The proposed approach represents an innovation in terms of resolution, extension of the analysed area with respect to previous regional landslide susceptibility and hazard zonation models applied in Central Asia (e.g., Nadim et al., 2006; Havenith et al., 2015b; Stanley and Kirshbaum, 2017; Pittore et al., 2018; World Bank, 2020). For the studied area the landslide susceptibility distribution in the area covered by elements at risk, such as roads, railways, and buildings, was also assessed (Scaini et al., in prep).

## 2. Study area

Geographically, Central Asia is a vast and diverse region including high mountain chains, deserts, and steppes (Fig. 1). A large portion of the Central Asia countries, especially the southern and eastern parts of the region, are occupied by the mountainous areas of the Djungaria, Tien Shan, Pamir, Kopetdag, and small part of Western Altaj, with peaks above 7,000 m a.s.l (Strom, 2010). These intraplate mountain systems formed in the Cenozoic between the Tarim Basin and the Kazakh Shield, as a result of the India-Asian collision (Molnar and Tapponier 1975, Abdrakhmatov et al., 1996; 2003; Zubovich et al., 2010, Ullah et al., 2015). This work is focused in the most inner part of Central Asia, represented by the territories of Turkmenistan, Kazakhstan, Kyrgyz Republic, Uzbekistan, and Tajikistan. Active mountain building started in the Oligocene (Chedia 1980) or even later (Abdrakhmatov et al. 1996), forming a complex system of basement folds disrupted by numerous thrusts and reverse faults with significant amount of lateral offset (Delvaux et al. 2001). Several regional fault zones are aligned along large parts of the mountain belts, others cross the orogen in a NW-SE direction, e.g., the Talas-Fergana fault, which forms a distinct boundary between the western and central Tien Shan (Trifonov et al. 1992) (Fig. 2).

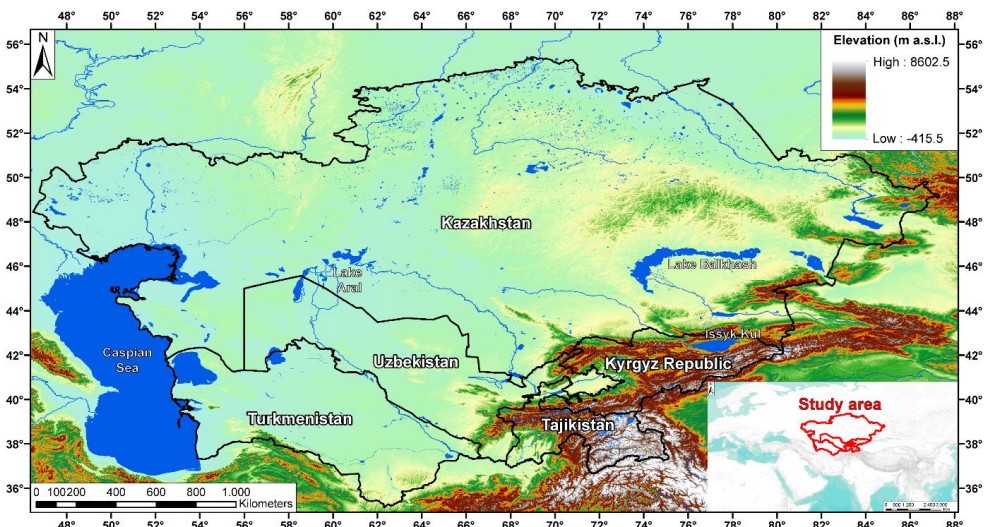

**Figure 1**. **Study area geographical-geomorphological setting.** Lakes' polygons from Schiavina et al., 2022, while MERIT DEM (Yamazaki et al. 2017) was used as topographic base.

Mountain ridges, formed mainly by palaeozoic crystalline rocks, are separated by wide lenticular or narrow, linear intermountain depressions, containing Neogene and Quaternary deposits, mainly sandstone, siltstone with gypsum interbeds, and conglomerates (Strom and Abdrakhmatov, 2017). Mesozoic and Paleogene deposits are typical of the foothill areas. Almost every ridge, especially in the Tien Shan, corresponds to a neotectonic anticline, and most of the main river valleys follow intermontane tectonic depressions, which are linked by narrow deep gorges up to 1-2 km deep (Strom and Abdrakhmatov, 2018). These mountain systems are the sources of most of Central Asia rivers, which, being fed by glaciers, snowmelt water and rain, have deeply incised valleys.



Such extreme topography along with complex geological structure, active tectonics and high seismicity determine
important landslide predisposing factors, making landslides the third most prevalent natural hazard in Central Asia,
following earthquakes and floods (CAC DRMI, 2009; Havenit et al, 2017).

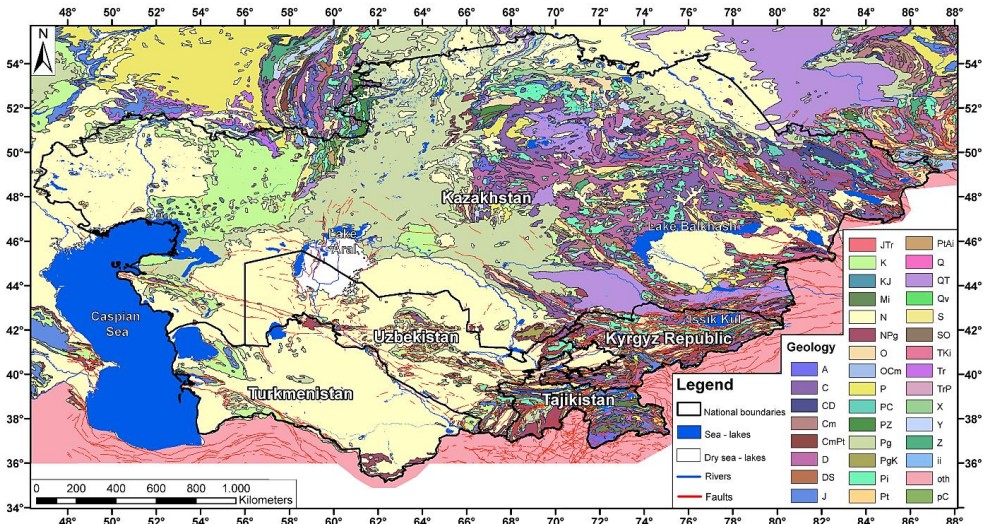

**Figure 2. Geological map of the study area.** Geological formation data from United States Geological Survey
(see Persits et al., 1997 for the legend), including faults from the AFEAD (Active Faults of Eurasia) database
(Styron and Pagani, 2020).
**2.1 Landslide types in Central Asia**
According to the international Cruden and Varnes 1996 classification, landslides phenomena in Central Asia
include rockslides/rock avalanches, rotational/translational slides and mud/debris flows (often involving loess),
which are triggered by natural events such as earthquakes, floods, rainfall and snowmelt (Behling et al., 2014;
2016; Golovko, 2015; Havenith et al., 2006a,b, 2015a, b; Kalmetieva et al., 2009; Saponaro et al., 2014; 2015;
Strom and Abdrakhmatov, 2017; 2018). Glacial lakes outburst flood phenomena, caused by the breech of natural
glacial dams, often result in large scale catastrophic mud/debris flows. In Central Asia, landslides more often occur
in the loess zone of contact with other rocks, on clay interlayers of the Mesozoic and Cenozoic age, reaching a
volume from tens of thousands up to 15-40 $*10^6$m$^3$ (Juliev et al., 2017). Seismically triggered landslides are very
common in tectonically active mountain regions, such as Tien Shan and Pamir (Sternberg et al., 2006; Hong et al.,
2007; Juliev et al., 2017). According to the literature background, most of the large mapped mass movements
(especially those with a volume of more than $10^6$ m$^3$) were triggered generally by major (also prehistoric)
earthquakes, possibly in combination with climatic factors, namely snowmelt and heavy rainfall (Havenith et al.,
2003; Strom and Korup, 2006; Strom, 2010; Schlögel et al., 2011; Strom and Abdrakhmatov 2017, 2018; Havenith
et al., 2015a; 2016; Behling et al., 2014; 2016; Piroton et al., 2020). Furthermore, in the past few decades, the
number and intensity of landslides have grown owing to climate change and the increase of the anthropic pressure,
due to several factors such as the uncontrolled land and water use, the rising of the water tables (often induced by


the increase of irrigation; Ishihara et al., 1990), mining, and excavation activities (Pollner et al., 2010; Thurman,

147 2011).

**2.2 Large Rockslides and natural dams**

Numerous rockslides have occurred in the mountains producing hazardous natural phenomena such as long runout
rock avalanches (Fig. 3) and dammed lakes, more than 100 of which still store water (Strom, 2010). These mainly
involve the Palaeozoic magmatic and metamorphic crystalline bedrock, but also the sandstone and limestone
formations. Although according to Strom, 2010, many of the existing dammed lakes should be considered as stable,
catastrophic outburst floods that occurred in the 20$^{th}$ century, emphasize high potential hazard of landslide natural
blockages. Havenith et al., 2015a report a catalogue of large to giant landslides (having volumes exceeding $>10^7$
m$^3$) in the Tien Shan area, showing several information such as location, time of occurrence, volumes, and
thickness. Regarding the volumes of these rockslides, these range from $50*10^3$ m$^3$ to 10 km$^3$ (Strom and Korup,
2006; Strom and Abdrakhmatov, 2018). Many of these phenomena, though not all, were triggered by earthquakes
with M > 6 and have dammed a river valley (some of the dams have been naturally or artificially breached)

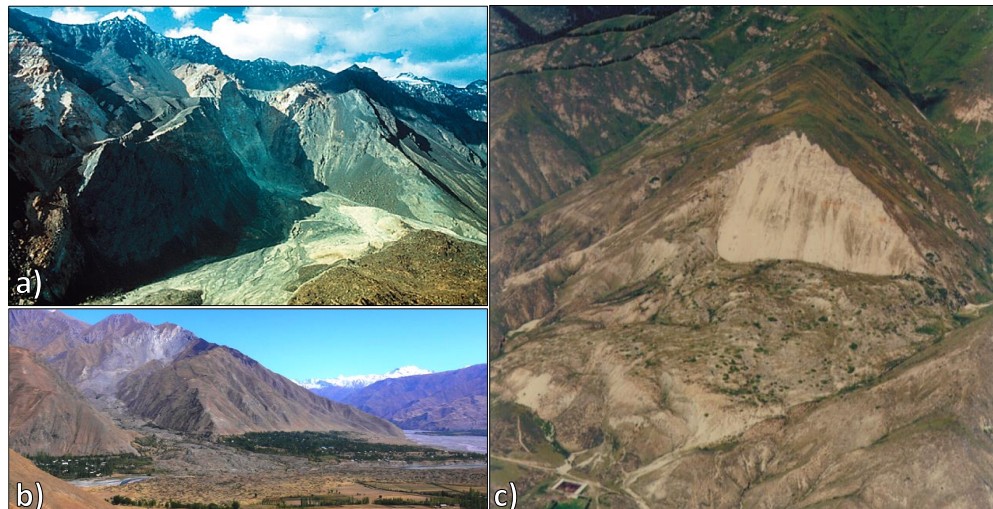

**Figure 3. Examples of large rockslide features in Central Asia.** Helicopter view of the Usoi landslide scarp,
triggered by the 1911 earthquake, Tajikistan (a) (after Strom, 2010); Khait rock avalanche (b) (after Havenith et
al., 2015a); helicopter view of Ananevo landslides (c) (after Havenith et al., 2015a).

**2.3. Landslide in soft rocks and loose deposits**

Rotational landslides mostly occur in loose unconsolidated Quaternary deposits, and in soft and semi-hard rock
layers in Mesozoic-Cenozoic sediments, represented mainly by layers of clays, claystones, siltstones, sandstones,
marls, limestone, gypsum, and conglomerates, with intercalated clays (Roessner et al., 2004; Kalmetieva et al.,
2009) (Fig. 4). These phenomena can create river dams, but they rarely are long-living dams, since usually they
are small and their bodies are eroded quickly even if they block a river channel (Strom and Korup, 2006). The
loess landslides occur quite regularly (on a yearly basis) in the regions presenting an almost continuous and locally
very thick (>20 m) cover of this material, generally at mid-mountain altitude (900 - 2,300 m) and mainly along the
border of the Fergana Basin (Kyrgyz Republic, Uzbekistan, and Tajikistan), and on the southern border of the Tien
Shan in Tajikistan (Fig. 4).

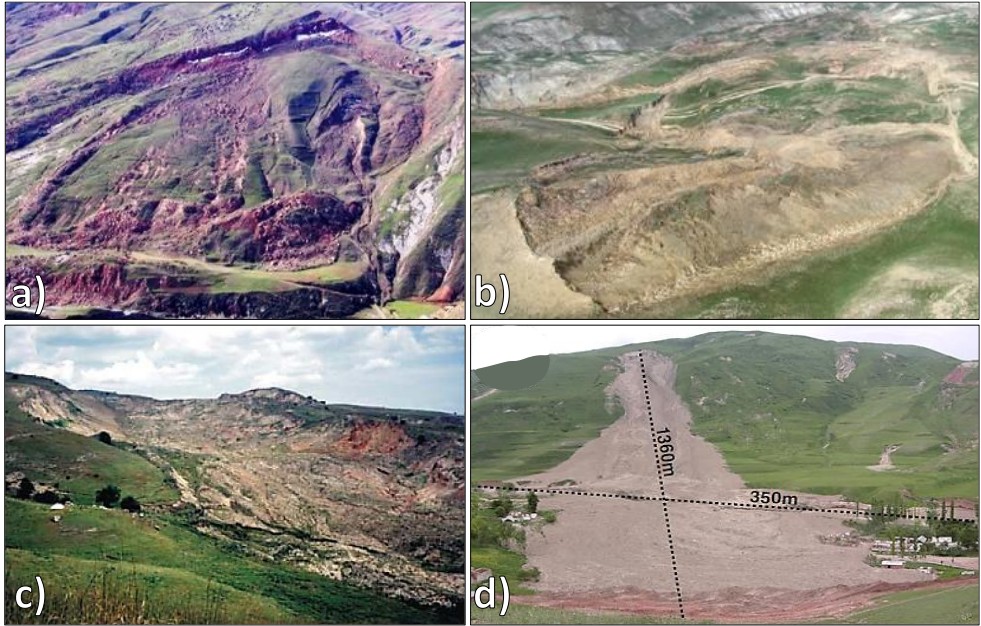


**Figure 4. Examples of landslides in soft rocks-loose deposits.** Picture the Kamar landslide (a) and the Beshbulak
landslide (b) (after Niyazov and Nurtaev, 2013). Examples of loess slides and mixed loess—soft landslides in NE
Fergana valley: Kochkor-Ata landslide failure in spring 1994 (c) (after Roessner et al., 2005); Field photo of the
Kainama landslide (d) (after Behling et al., 2016).
Loess flow landslides and debris flows, involving the eluvial slope cover, represent a relevant hazardous
phenomenon in the mountainous regions of Kazakhstan, in the area of Almaty, near the southern border with
Kyrgyz Republic, in the Altai area (Medeu and Blagovechshenskiy, 2016), around the Fergana Basin, all along
the border between Tajikistan and Kyrgyz Republic and around the Tajik Depression. Landslides occurring in
Quaternary loess units of up to 50 meters thick are characterized by very rapid avalanche-like mass movements,
which can reach several meters per second (often represent a combination of rotational slide and dry flow resulting
in long runout zones; World Bank, 2008). Typically, pure loess landslides have a volume of hundreds up to one
million cubic meters and appear as clusters (Roessner et al., 2005). From the recent history it appears that pure (or
quasi-pure) loess slides and flows are particularly dangerous because of their high velocity and long runout which,
in turn, can generate a great destructive power and more severe disasters than other types of mass movements of
similar size (Havenith et al., 2015a; Behling et al., 2014; 2016). If failure also affects underlying materials (mostly
Mesozoic and Cenozoic soft rocks), the volume of these mixed slides can exceed $10 \times 10^6$ m$^3$. These kinds of
landslides are particularly deadly and can be triggered by a combination of long-term slope destabilization factors
(e.g., rainfall and snowmelt) and short-term triggers (e.g., seismic shocks).




Even though earthquake-triggered loess slides and flows are far less frequent than rainfall triggered ones, they
caused much larger disasters in recent history, such as those triggered, respectively, by the July 1949 Khait and
the January 1989 Gissar earthquakes. The number of active debris flow basins in Kazakhstan is over 300 with
registered cases of more than 600 debris flows of different genesis (80% of which are represented by heavy rainfall-
triggered debris flows, while the glacial debris flows make up about 15% of the total) (Yaning, C., 1992).
**3 Materials and Methods**
*3.1 Landslide databases*
To implement the adopted susceptibility models the largest, most accurate, and updated landslide inventories were
used (Fig. 5). These were compiled by means of decades of field surveys, remote sensing and geophysical analysis
in the study area.
Hereafter we report their description in detail:
• The "Rockslides and Rock Avalanches of Central Asia" (Strom and Abdrakhmatov, 2018): a large inventory
including over 1000 polygons of large-scale (>=1 Mm$^3$) rockslides and rock avalanches, covering central
Asian countries (except for Turkmenistan and Altai) plus Chinese Tien Shan and Pamir, and Afghan
Badakhshan. Compiled in decades of field work and analysis of aerial/satellite imaging, it also comprises
information on landslide morphometric parameters (runout, area), and 126 polygons on possible landslide
bodies, dammed lakes, and head-scarps. Quantitative characteristics (area, volume, runout, etc.) for about 600
cases are provided as well.

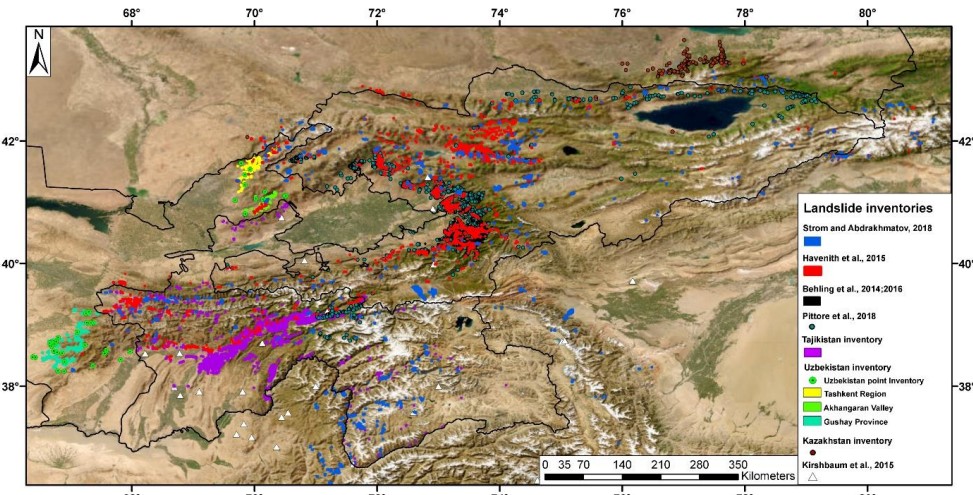

**Figure 5**. **Map of the adopted landslide inventory maps.** Basemap source: Esri, Maxar, Earthstar Geographics,
and the GIS User Community.
• The "Tien Shan landslide inventory" (Havenith et al., 2015a): represents the largest inventory in the study
area. Compiled by means of field surveys, remote sensing data interpretation and geophysical surveys, it
comprises the rockslides of the previous inventory together with other smaller landslides in soft sediments
(Havenith et al. 2006a; Schlögel et al., 2011) for a total of 3,462 landslides polygons, also including
information on landslide length and area.
• The "Multi-temporal landslide inventory for a study area in Southern Kyrgyz Republic derived from RapidEye
satellite time series data (2009 – 2013)" (Behling et al., 2014; 2016; 2020), is a semi-automated spatiotemporal
landslide inventory for the period from 1986 to 2013, covering a 2,500 km$^2$ in the Fergana valley rim in
southern Kyrgyz Republic. This inventory includes 1,582 landslide polygons mapped from multi-sensor
optical satellite time series data, together with information on spatiotemporal landslide activity patterns (area
and year of trigger).
• "The EMCA landslide catalogue Central Asia" (Pittore et al., 2018), including 3,130 points, which covers
mostly western and northern Kyrgyz Republic as well as Tajikistan's Region of Republican Subordination.
The catalogue is a summary (point locations) of the documented landslides between 1954 and 2009
(Kalmetieva et al., 2009), which are collected by the Central Asian Institute for Applied Geosciences through
geological surveys (field campaigns) on single sites close to urban areas.
• The "Tajikistan landslide database" provided by the Institute of Water problems, Hydropower, Engineering
and Ecology of Tajikistan (IWPHE), which includes 2,710 landslide polygons and 114 landslide-prone areas
(with information on length and area).
• The landslide inventory provided by the Institute of Seismology of the Academy of Science of Uzbekistan
(ISASUZ), which covers the Tashkent province. It comprises a point inventory (including location, type,
volume, length, and date of triggering; Nyazov R.A. 2020) and a polygon inventory digitized for this project
from the maps in Juliev et al., 2017 (including a total 345 landslide polygons).
• The landslide inventory, provided by the LLP "Institute of Seismology" of the Science Committee of the
Republic of Kazakhstan, covering mainly the Tien Shan area at the border with Kyrgyz Republic, and small
part of the western Altai, including 254 point-objects with information on type, area/volume, triggering date.
• Part of the "Global Landslide Catalogue (GLC)" (Kirshbaum et al., 2015), which covers Kyrgyz Republic and
Tajikistan, including 15 landslide point with a description on landslide size/type and triggering date/factor.
The GLC was compiled since 2007 at NASA Goddard Space Flight Centre NASA and considers all types of
mass movements triggered by rainfall, which have been reported in the media, disaster databases, scientific
reports, or other sources.
**3.2 Random Forest (RF) model**
To generate the landslide susceptibility maps in this work, the Random Forest model (RF) was used. The RF is a
nonparametric and multivariate machine learning technique, which was proposed by Breiman (2001), and first
used in landslide susceptibility analysis by Brenning (2005). Since then, it has rapidly gained widespread
consolidation through many research and case studies, as it is considered a relatively powerful approach in
classification, regression, and unsupervised learning (Lagomarsino et al., 2017). Among the advantages of using
the RF algorithm, there is the possibility of using numerical and categorical variables at the same time, without
assumption on the statistical distribution of their values. Furthermore, RF is acknowledged to be capable of
handling implicitly the multicollinearity of variables, identifying the uninfluential (or the detrimental) ones




(Breiman, 2001; Brenning, 2005). The RF also automatically performs a validation by building a Receiver
Operating Characteristic Curve (ROC Curve) and calculates the relative Area Under the Curve (AUC). AUC is
widely used as a quantitative indicator for the predictive effectiveness of susceptibility models: it can range from
0.5 (completely random predictions) to 1.0. This model, by means of the bootstrapping technique, also calculates
the Out-of-Bag Error (OOBE) for each variable. This parameter measures the relative error that would be
committed if a given variable is excluded from the RF classifier. OOBE can be used to assess the relative
importance of each independent variable, thus representing a powerful tool to interpret the results and to rank the
variables according to their importance (Catani et al., 2013). RF contains a series of binary tree predictors, which
are generated by using a random selection of the input data (the independent variables which in LSM studies, are
a set of physical parameters representing the predisposing factors), in order to split each binary node (yes/no), and
to perform a classification of the target dependent variable (in LSM studies, the presence or absence of landslides).
Some of the observations are used for internal testing to evaluate the predictive capability of each predictor tree.
This information is used to iterate the procedure hundreds of times by growing other random trees (hence the name
"Random Forest"), and to iteratively adjust the prediction effectiveness. Once the best predictor tree is identified,
it is applied to the whole study area, to define the LSM. Another important key point of RF is that it has a great
predictive performance and runs fast by summarizing many classification trees and this is particularly useful when
dealing with large amounts of data.
**3.3. Selection of independent variables**
As independent variables, twenty "basic parameters" were selected in all 5 countries, based on the available data
and according to the ones most widely adopted in literature (Catani et al., 2013; Reichenbach et al., 2018). Many
of these are DEM-derived products (e.g., elevation, aspect, slope, slope curvature, flow accumulation, Stream
Power Index, Topographic Wetness Index, Topographic Position Index). It must be considered that the resolution
of the susceptibility maps depends on the resolution of the input data. Therefore, it was decided to use pixels
corresponding to the MERIT DEM (Yamazaki et al. 2017) resolution (3" – ca. 90 m at equator and ca 70 m at 40°
latitude). In addition, the DEM itself was used as a reference map, so that the other parameters were processed to
have a perfect overlapping. Therefore, the resulting landslide susceptibility maps will also be perfectly overlapping
to it. The variables such as lithology and soil type were rasterized with this resolution by choosing the most
frequent value in a reference window. The twenty "basic parameters" used are listed below, including a brief
description:
• MERIT DEM and DEM-derived products: Aspect, Slope Gradient, Total Curvature, Profile Curvature, Planar
Curvature, Flow Accumulation, Topographic Wetness Index (TWI), Stream Power Index (SPI), Topographic
Position Index (TPI).
• Lithology, derived from the geological map of the former Soviet Union made by the USGS (Persits et al. 1997).
• Soil type map from the DSMW database (Copernicus land use; https://land.copernicus.eu/).
• Distance from Faults: it is minimum distance, in meters, between each landslide and the nearest fault. The fault
database is derived from the AFEAD catalogue (Styron and Pagani, 2020) and was modified after Poggi et al.,
a (in prep.).



• Distance from Roads: it is minimum distance, in meters, between each landslide and the nearest road. The roads

database is derived from Scaini et al., (in prep.).

• Distance from Rivers: it is minimum distance, in meters, between each landslide and the nearest river. The river

network database is derived from Coccia et al., (in prep.).

• Distance from Hypocentres: it is minimum distance, in meters, between each landslide and the nearest

earthquake hypocentre with a magnitude greater than 6.5 (following the methodology adopted by Havenith et

al., 2015a). The Hypocentre database was provided by Poggi et al., a (in prep.).

• Peak Ground Acceleration (PGA): 4 kind of PGA maps according to different return times (475 and 1000 years)

and different materials (soil layers and bedrock) to which it refers were created (Poggi et al. b, in prep.).

In addition to these "basic parameters", in this study it was decided to use five parameters related to the propensity
of the territory to be affected by precipitation (Fig. 6). These parameters were obtained from the ERA5 database
(www.ecmwf.int/en/forecasts/datasets/reanalysis-datasets/era5).

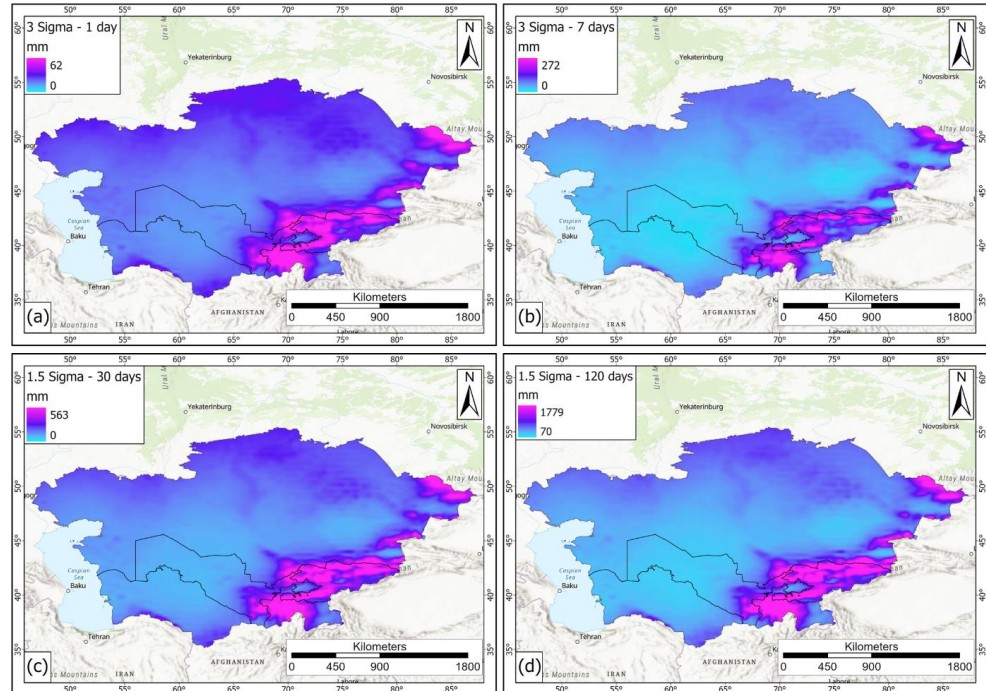


**Figure 6. Rainfall maps from the ERA5 database** (www.ecmwf.int/en/forecasts/datasets/reanalysis-
datasets/era5). (a) rainfall amounts corresponding to 3 standard deviations for 1-day rainfall; (b) rainfall amounts
corresponding to 3 standard deviations for 7-days rainfall; (c) rainfall amounts corresponding to 1.5 standard
deviations for 30-days rainfall; (d) rainfall amounts corresponding to 1.5 standard deviations for 120-days rainfall.
Basemap source: Esri, USGS, NOAA.
These data span from 1981 to 2020, have a 1-hour temporal resolution (summarized to daily resolution for this
work) and a spatial resolution 0.25°. The first parameter is the Mean Annual Precipitation (MAP) map, where, for
each pixel, the mean annual precipitation was calculated (Fig. 6). Other maps (named Sigma maps) have been
calculated by the spatialization of the approach described in Martelloni et al (2011). In detail, for each rain gauge
(represented by the pixels of ERA5 maps in this work) the rain values corresponding to a given standard deviation
for several cumulative intervals are defined (e.g., the rain values corresponding to 2 standard deviations of the
distribution of 3-days cumulative rainfall):
• Sigma 1.5 – 120 days: rainfall values corresponding to 1.5 standard deviations of the 120-days cumulative
rainfall They range from 70 mm to 1778.8 mm (Fig. 6a).
• Sigma 1.5 - 30 days: rainfall values corresponding to 1.5 standard deviations of the 30-days cumulative rainfall.
They range from 0 mm to 563.1 mm (Fig. 6b).
• Sigma 3 - 1 days: rainfall values corresponding to 3 standard deviations of daily cumulative rainfall. They range
from 0 mm to 62.2 mm (Fig. 6c).
• Sigma 3 - 7 days: rainfall values corresponding to 3 standard deviations of the 3-days cumulative rainfall. They
range from 0 mm to 271.9 mm (Fig. 6d).
The sigma parameters represent the probability of having a given rainfall amount over a defined time interval. In
this work, four intervals were selected (1, 7, 30 and 120 days) to consider both short and long rain events, that can
lead to the triggering of surficial or deep-seated landslides, respectively. For 1 and 7 days the maps of the rainfall
values corresponding to 3 standard deviations over the mean rainfall were selected, to verify if short and very
intense rainfall (with a very low probability of occurrence) could influence the slope stability in the study area.
Regarding the 30-days and 120-days interval, rainfall values corresponding to 1.5 standard deviation were
calculated, in order to assess the influence of longer and less intense rainfalls over slope stability.
**3.4. Model optimization**
The LSM was defined using the whole study area, instead of processing each country individually. This choice
allowed to overcome the boundary effects associated with the use of independent countries. In addition, a buffer
of 10 km was considered around the whole area, to avoid deformation due to boundary effects. These choices were
helpful in reducing distortions and improving the quality of the results, but also led to a huge amount of data to be
processed. Since the same resolution of the DEM was used for susceptibility assessment, the whole area was
divided into about $1.07 * 10^9$ cells and for each cell 26 condition factors and 1 dependent variable were defined;
this led to about $2.89 * 10^{10}$ data to be processed. In order to reduce the processing time and avoid computational
problems due to the huge amount of data and to the width of the study area, large flat areas were filtered and not
considered in the modelling process, since landslides generally take place along slopes (some exceptions to this
statement in the area are represented by landslide around the flat Caspian Sea area (Pánek et al., 2016). For
Turkmenistan no landslide database was available, so it was decided to train and test the model only with the other
4 countries, to obtain the best predictor model for the available data. The trained model has then been applied to
the whole study area, including Turkmenistan, to define the LSM. Regarding the dependent variables, the
landslides inventory was created by merging the data described in section 3.1. As a result, this landslide data was





quite heterogenous, hence an initial control and homogenization phase was necessary. In this framework the
landslide data were checked to verify the presence of overlapping polygons or topological errors, which were
removed. Since some landslide inventories were composed solely by points, these were mapped only as a
"landslide points", a 100 m buffer was created around them, in order to include them in the model. However, when
the points refer to large landslides, which are frequent in the study area, it is possible that part of the body of these
landslides is still outside the perimeter achieved with the buffer. To avoid classifying these areas as non-landslide
points, it was decided to create an additional buffer of 1 km around points, used as a mask where the non-landslide
points were not to be selected. This process reduced the probability of pixels misclassification (e.g., landslide
points considered as non-landslide points) during the training of the model. All the points inside the 1-km buffer
were only considered during the model application, as well as point from Turkmenistan. Some landslide-prone
areas were also present in the input inventories; since these were not real landslides but 'landslide-prone zones',
these areas were not used to train the susceptibility model but were used in the validation of the results. This
optimization procedure allowed to define an input dataset of $1.08*10^8$ points (along with 27 variables for each
point) to be used to define the susceptibility model. A further optimization of the model was performed by the
evaluation of the out of bag classification error, i.e., the variation of the misclassification probability with the
number of grown classification trees. The classification error initially reduces with the increasing of classification
trees, then it turns to be stable, so the definition of the optimal number of classification trees is required to avoid
the use of an overgrown forest with an excessive number of trees (hence with high computational load and time)
and without any advantage for the model (Fig. 7).
**3.5. Model training**
Once all the data were prepared and organized, the algorithm to create the landslide susceptibility maps was
developed. A crucial step in LSM analysis is the approach used to sample the variables to train and validate the
model. As in any other statistical procedures, the size of the dataset influences the results, therefore the higher the
number of samples to perform the statistical calibration/validation of the model, the more reliable are the obtained
results. To avoid a generalized hazard overestimation, Catani et al. (2013) demonstrated that a random sampling
improves the predictive capability of the map, and that the susceptibility model should also be trained/validated
with respect to information about non-landslide locations. Regarding the proportion between the calibration and
validation dataset samples, it is common practice to split them according to a 70/30 ratio. Therefore, using ESRI
ArcGIS Pro software, all the variables were sampled pixel by pixel, after which, with the Matlab software, from
the total of the sampled points, all the points within a landslide and a same amount of randomly chosen non-
landslide points were extracted. This input dataset was divided into two parts, 70% of the data (calibration dataset)
was used for the training phase, and the remaining 30% (validation dataset) for the testing phase. The selection
and division were randomly repeated 5 times, in order to assess the stability of the model to the variation of the
training and testing datasets, hence, to verify the absence of overfitting issues. Each one of these datasets was
created to be equally composed by pixel within a known landslide and pixel outside a landslide. All these data
were then used to train and test the algorithm created to predict the landslide susceptibility of the whole area.

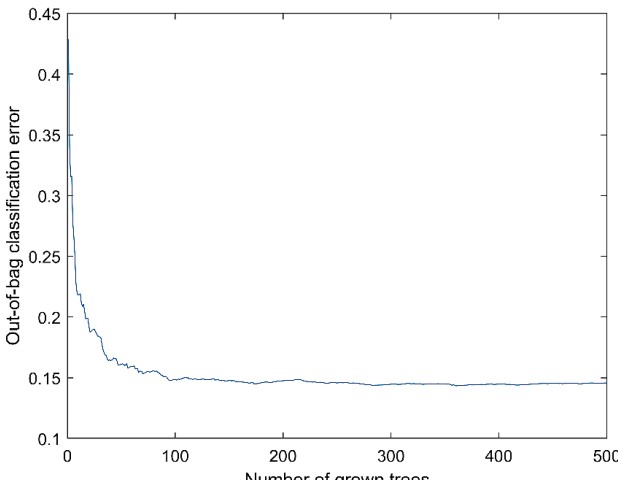


**Figure 7. Example of out of bag classification error.** The error is stable using 100 or more trees.

The best predictor model identified in the training phases was then applied to all the available data (also for
Turkmenistan and for the 1-km buffer area around the point-object landslides) for the development of the
susceptibility map on the whole Central Asia area. The results obtained from the application of the aforementioned
methodology are the susceptibility map, the ROC (Receiver Operating Characteristic) curves with their AUC (Area
Under the Curve) values, and the histogram of the importance of variables. ROC and AUC are used to verify the
quality of the landslide susceptibility model, both by graphical and analytical approach. Due to the high volume
of data, their variety, values, and heterogeneity a specific algorithm was created for this work, that was set to be
able to perform several activities:
• Reading and properly formatting the input data and then dividing them between independent and
dependent variables.
• Automatically and randomly selecting locations associated with landslides or outside landslides to create
the training and test datasets.
• Identifying the best predictor and evaluating its performances by the calculation of the misclassification
probability of the values calculated by the model.
• Evaluating the overall performances of the model by the mean of ROC and AUC.
• Identifying the importance of the parameters in landslide susceptibility.
• Applying the model to the whole study area, calculate the probability of classification (landslide or non-
landslide) of each pixel and extraction of the final map in raster format.
The algorithm was set to work in classification mode, e.g., for each pixel a value (1 or 0) is assigned to identify
the presence or absence of a landslides (dependent variable), along with the values of the independent variables.
Using these data, the RF model identifies the best association of independent variables linked to presence or
absence of landslides (landslide susceptibility prediction model).



The prediction model is then applied to all the pixels of the investigated area, and the probability of each pixel to
be classified as landslide (or non-landslide) pixel is evaluated. These probability values are those used to create
the landslide susceptibility maps. It must be noticed that the landslide inventories adopted to train the RF rarely
reported the type of landslide, so the LSMs must be considered not related to a specific type of landslide.
**3.6. Model validation**
To verify the quality of the susceptibility models, beside the AUC value previously reported, a confusion matrix
for the four countries where the model was trained was created (Fig. 8). In each matrix the predicted landslide
classes are compared with the ground truth to verify the presence of significant misclassification error. In all the
matrix the value 1 represent the presence of landslide, the value 0 represents the absence of landslides; the numbers
in each cell represent the number of pixels classified in that combination of 0 and 1, according to this scheme (the
first number represent the predicted class, the second number the ground truth):
- 0-0 (True negative): pixels outside any landslides are correctly identified as no-landslide pixels by the
model.
- 1-1 (True positive): pixels inside a landslide are correctly identified as landslide pixels by the model.
- 0-1 (False negative): pixels inside a landslide are wrongly identified as no-landslide pixels by the model.
- 1-0 (False positive): pixels outside any landslides are wrongly identified as landslide pixels by the model.

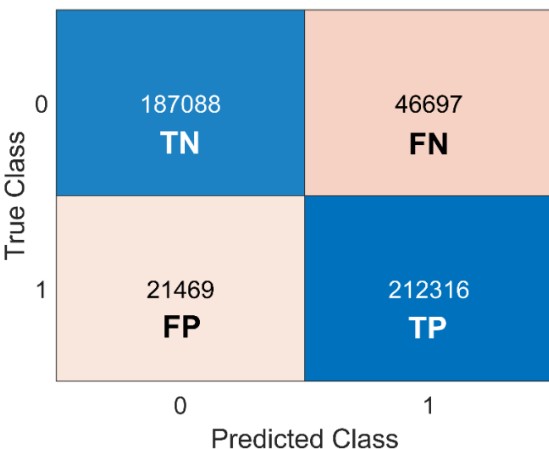


**Figure 8. Confusion matrix for the four countries where the model was trained.**
The 0-0 and 1-1 combinations represent well classified pixels (blue cells in Fig. 8), while 0-1 and 1-0 represent
misclassification error (light red cells in Fig. 8). Since this matrix needs some ground-truth parameters (True
classes), it can be applied only where the presence or absence of landslides is known. For this reason, in this work,
this matrix was calculated considering only the test dataset. A further control of the results was made using the
areas prone to landslides identified in the used landslide inventories.

**3.7. Landslide susceptibility and elements at risk**

The susceptibility map of the study area was intersected with the elements at risk, consisting of roads-railways, population, to analyse the landslide susceptibility distribution in the area covered by elements at risk. The database of element at risk was provided by Scaini et al., in prep. In order to perform the analysis several approaches were defined based on the different types of elements at risk. the population and buildings data were based on a grid with a spatial resolution of 1km$^2$, defining for each cell the number of inhabitants, the number of different types of buildings (residential, commercial, industrial, education and healthcare), and the mean susceptibility class by means of spatial statistics between input databases (population-buildings data and susceptibility map). The results carried out from the spatial statistics allowed to assess the people and buildings distribution within each susceptibility class. On the contrary, the linear elements (roads and railways) were divided in segments with 1-km in length, and buffered, setting a distance parameter equal to 100 m. After this preliminary process, the spatial statistics with the landslide susceptibility have been carried out.

**4. Results**

**4.1 Susceptibility map**

In the map presented in the following Figures 9 and 10, the susceptibility values, ranging from 0 to 1, were classified into five classes (Table 2). Here the corresponding extension and percentage of the study area are also reported, showing that the most frequent susceptibility class for the whole study area is the null class (=87.8%; landslides generally don't occur in flat areas), followed by low and medium classes. Only the 4% of the central Asian territory is represented by areas with high and very high landslide susceptibility (Table 2).

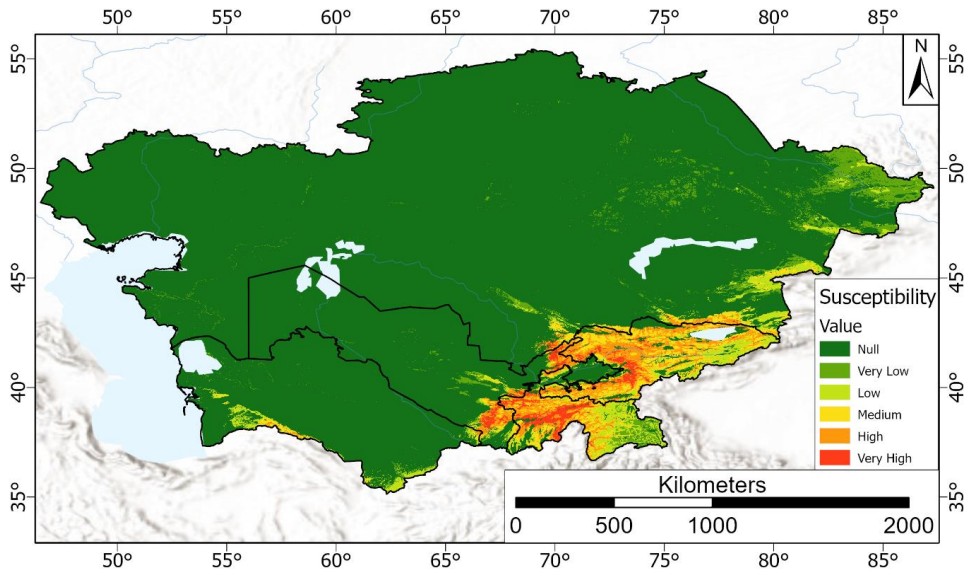

**Figure 9. Landslide susceptibility map of Central Asia.** Basemap source: Esri, USGS, NOAA.

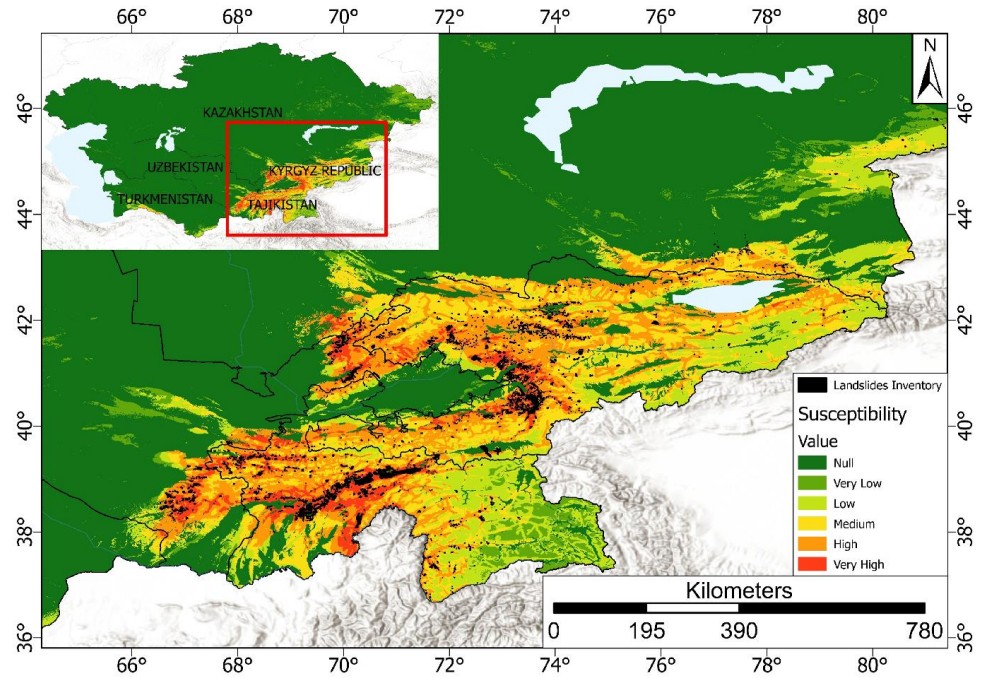

**Figure 10. Detail of the landslide susceptibility map with the overlapping landslide polygons (in black).** On the top left the detailed area with respect to the central Asian territory. Basemap source: Esri, USGS, NOAA.

**Table 2.** Landslide susceptibility class intervals, corresponding area, and percentage with respect to CA.

| Susceptibility class | Landslide spatial probability interval | Corresponding area (km²) | Corresponding percentage of CA (%) |
|---|---|---|---|
| Null | 0 - 0.05 | 2,889,481.2 | 87.8 |
| Very Low | 0.05 - 0.25 | 94,674.7 | 2.9 |
| Low | 0.25 - 0.35 | 85,294.1 | 2.6 |
| Medium | 0.35 - 0.45 | 87,528.5 | 2.7 |
| High | 0.45 - 0.6 | 99,689.8 | 3 |
| Very High | 0.6 - 1 | 31,436.4 | 1 |

In Fig. 11, the susceptibility maps of five selected areas are displayed. From these details it is possible to ascertain the high usefulness of the landslide susceptibility map realized by applying the Random Forest model, which, mainly based on the hydro-geomorphological properties, can establish the degree of susceptibility even in areas where there is no awareness of the predisposition to instability due to the absence of reported landslides. In particular:
- Fig. 11a shows the area north of the city of Denau, in the south-east of Uzbekistan, which is characterized by a high susceptibility, despite the almost total absence of mapped landslides.

- Fig.11b shows a detail of the city of Ura-Tube, in the North-West of Tajikistan, where there are not any known landslides, but a high susceptibility has been obtained in the surrounding mountain relief.

- In Fig. 11c there is a close-up on the city of Dushanbe, the capital of Tajikistan, where close to roads and inhabited centres a high landslides susceptibility is observed.

- The shores of Lake Issyk-Kul in the Kyrgyz Republic, shown in Fig. 11d, are generally flat areas, with a low or null landslide susceptibility but in the central zone.

- Finally, Fig. 11e shows a detail of the western area of the Kyrgyz Republic, where a high landslide susceptibility is observed along the slopes adjacent to the river network.

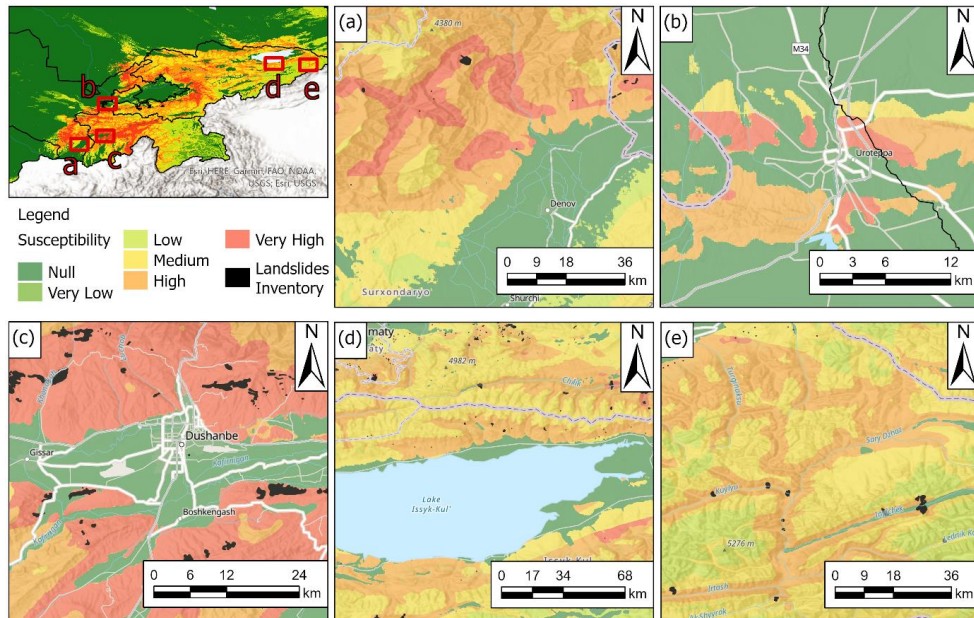

**Figure 11. Details of the landslide susceptibility map**. (a) the city of Denau, Uzbekistan; (b) the city of Ura-Tube, Tajikistan; (c) the city of Dushanbe, Kyrgyz Republic; (d) the Lake Issyk-Kul, Kyrgyz Republic; (e) the eastern area of the Kyrgyz Republic. Black poligons represent landslide areas from the adopted landslide inventories. Basemap source: Esri, USGS, NOAA.

### 4.2 The Fergana valley mountainous rim

The Fergana valley spreads across eastern Uzbekistan, southern Kyrgyz Republic, and northern Tajikistan (Fig. 12). It is one of the largest intermountain depressions in Central Asia, located between the mountain systems of the Chatkal-Kuraminsk ranges in the north and the Turkestan--Alai in the south. The two main rivers, the Naryn and the Kara Darya, flow into the valley and unite forming the Syr Darya. In this area landslides represent one of the major natural hazards due to their frequent (seasonal) occurrence across large areas: in fact, they are particularly

concentrated in a range of altitudes between 700 and 2000 m along the topographically rising rim below its
transition into higher mountainous terrain (Roessner et al., 2000; 2004; 2005; Behling et al., 2014; 2016). This
region is quite densely populated, and landslides lead almost every year to damage of settlements and infrastructure
and loss of human life (Schloegel et al., 2011; Piroton et al. 2020). In this area landslide activity is caused by
complex interactions between tectonic, geological, geomorphological and hydrometeorological factors (Havenith
et al., 2015a, b). In the Fergana valley rim mass movements are often characterized by deep and steep scarps,
mobilize weakly consolidated sediments of Tertiary or Quaternary age, including loess deposits (Piroton et al.,
2020). These kinds of landslides are particularly deadly, and can be triggered by a combination of long-term slope
destabilization factors (e.g., rainfall and snowmelt) and short-term triggers (Danneels et al., 2008). Slope landslide
susceptibility was analysed in this area using the previously mentioned methodologies. Fig. 12 shows the particular
about the landslide susceptibility map obtained for the Fergana Valley, while Fig. 13 reports the histogram of the
area occupied by each susceptibility class. It can be observed that the most frequent susceptibility class in the
Fergana Valley area is the Null class, which covers an area of about 20,743 km$^2$, that is 36% of the valley. The
Very Low and Low classes occupy respectively an area of 681 km$^2$ (1.2%) and 5,431 km2 (9.4%). The Medium
class instead extends for about 8,608 km2, namely the 15% of the total. The High class instead extends for about
16,395 km$^2$, that is 28.5% of the total and finally, the remaining 9.9% of the national territory, that is about 5,683
km$^2$, is classified in the Very High class.

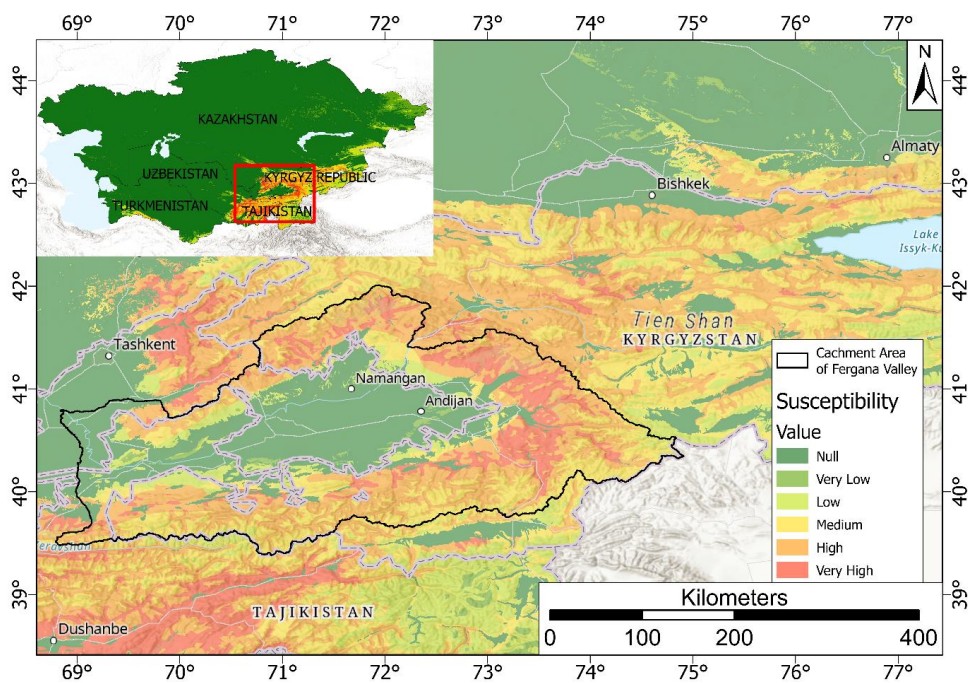


**Figure 12. Detail of the landslide susceptibility map obtained for the Fergana Valley.** Basemap source: Esri,
USGS, NOAA.

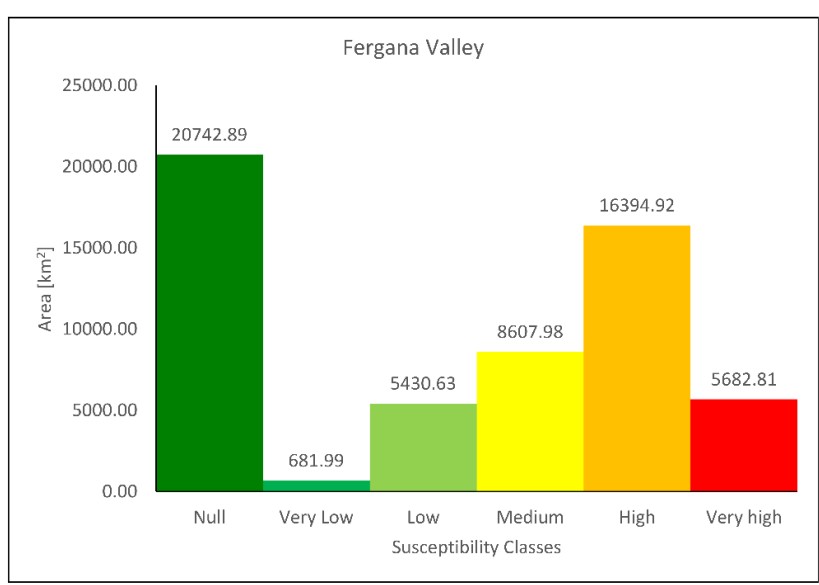


**Figure 13. Frequency histogram of susceptibility classes obtained for the Fergana Valley mountainous rim**.

On each bar the corresponding area in km$^2$ is reported.

**4.3 Trained model performances and conditioning factors relevance**

The RF was initially trained setting 1,000 trees to be growth. After the first run, the analysis of the out-of-bag error
revealed that misclassification probability reduced significantly with a forest of 150 trees and then reduced slightly
up to 500 trees, then it turned to be stable, so the optimal number of trees was set equal to 500 and used for all the
simulations. As described above, the model was run 5 times to verify its stability and the AUC values ranged from
0.93103 to 0.93144 (Fig. 14), with a mean value of 0.93122 and a standard deviation of 0.00015. The low variance
of the AUC values confirmed the stability of the model and its applicability to the whole area. As we can see in
the ranking of the susceptibility parameters, reported in Fig. 15, soil type, lithology, elevation, the distance from
roads and hypocentres plays a crucial role in landslide susceptibility, since they are the five most influencing
factors (for the four countries where the model was trained). Rainfall parameters are also important in the obtained
landslide susceptibility, in particularly the 1-day rainfall value that shows the highest importance among the
rainfall parameters. Also, the PGA maps are a relevant factor, while TWI and slope curvature are the less important
parameters. The average AUC value of the models is 0.93122, indicating their very good quality. Such high AUC
values can indicate the presence of overfitting issues, but this hypothesis can be discarded, since the random
variable resulted without any importance in landslide susceptibility (negative OOBE value).




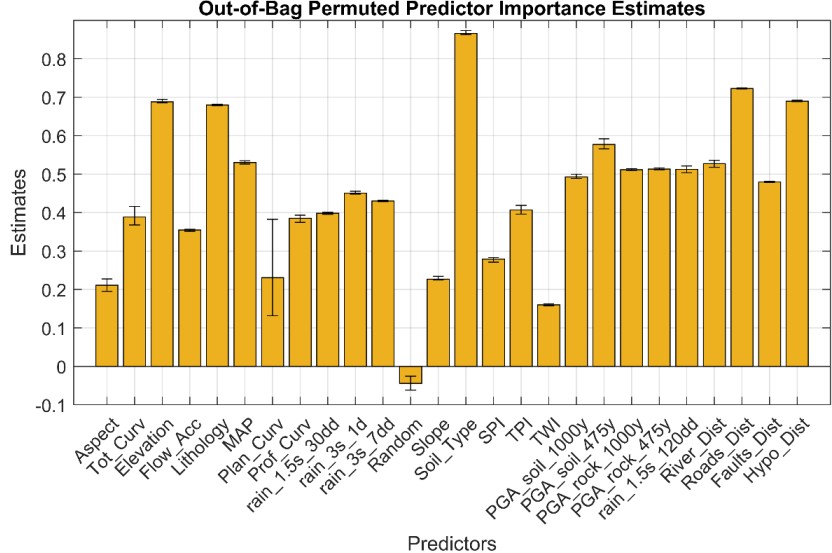


**Figure 14. ROC curve and relative AUC value for each model run.**


**Figure 15. Variable importance in landslide susceptibility for the four countries where the model was**

**trained**. From the 5 model runs, the results were averaged and displayed in this image, with the error bars showing

the maximum and the minimum value obtained.



**4.4 Landslide susceptibility and exposed elements**

Concerning the outcomes regarding buildings and population, they are represented by Table 3, in which for each susceptibility class the number of people and the number of different building types are reported, and in the bar diagram of (Fig. 16). The obtained results about roads and railways are reported in Table 4 and in Fig. 17.

**Table 3. Population and buildings distribution in each landslide susceptibility class.**

| Element at risk | Landslide susceptibility | | | | |
| --- | --- | --- | --- | --- | --- |
| | Null | Low | medium | high | very high |
| Population | 68,422,152 | 3,046,892 | 1,612,487 | 2,812,081 | 97,934 |
| Residential buildings | 8,769,270 | 319,776 | 245,754 | 386.628 | 12,753 |
| Commercial buildings | 2,196,037 | 103,745 | 68,187 | 68.232 | 3,410 |
| Industrial buildings | 705,352 | 14,776 | 6396 | 7024 | 110 |
| Education buildings | 42,472 | 1802 | 960 | 2102 | 96 |
| Healthcare buildings | 15,476 | 224 | 84 | 226 | 2 |

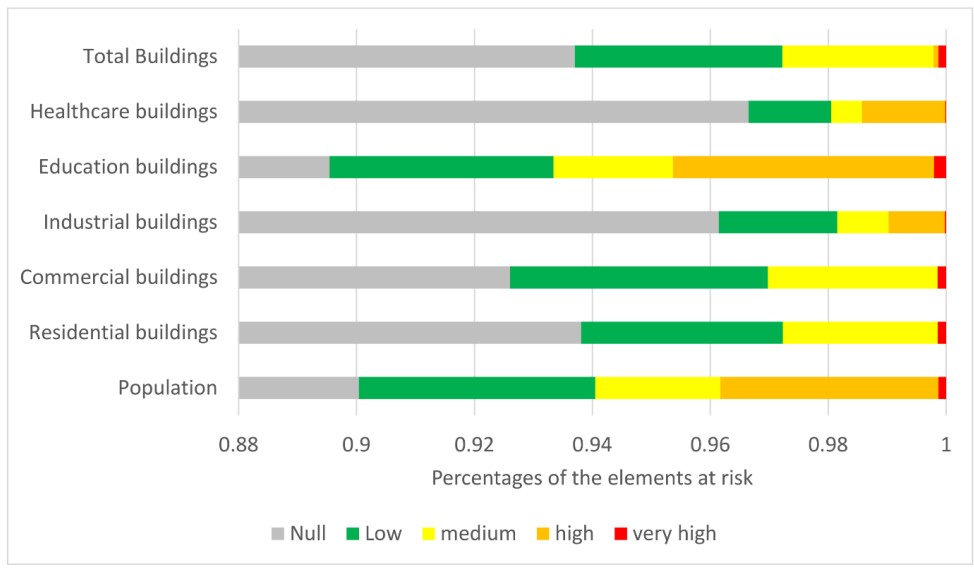

**Figure 16. Percentage of the Element at risk falling within landslide susceptibility classified areas.**





**Table 4. Distribution (corresponding km) of road and railway classes in landslide susceptibility classes.**

| | | Null | Low | Medium | High | Very High |
|---|---|---|---|---|---|---|
| Road Class | Primary | 15,000 | 646 | 368 | 873 | 26 |
| | Secondary | 28,773 | 911 | 589 | 1,173 | 30 |
| | Tertiary | 71,515 | 2,637 | 1,643 | 3,898 | 55 |
| | Trunk | 30,058 | 1,887 | 686 | 1,887 | 77 |
| | Motorway | 1,732 | / | / | / | / |
| Railway Class | High-Speed | 45,866 | 589 | 317 | 187 | 25 |
| | Conventional | 128 | 4 | 16 | 12 | / |

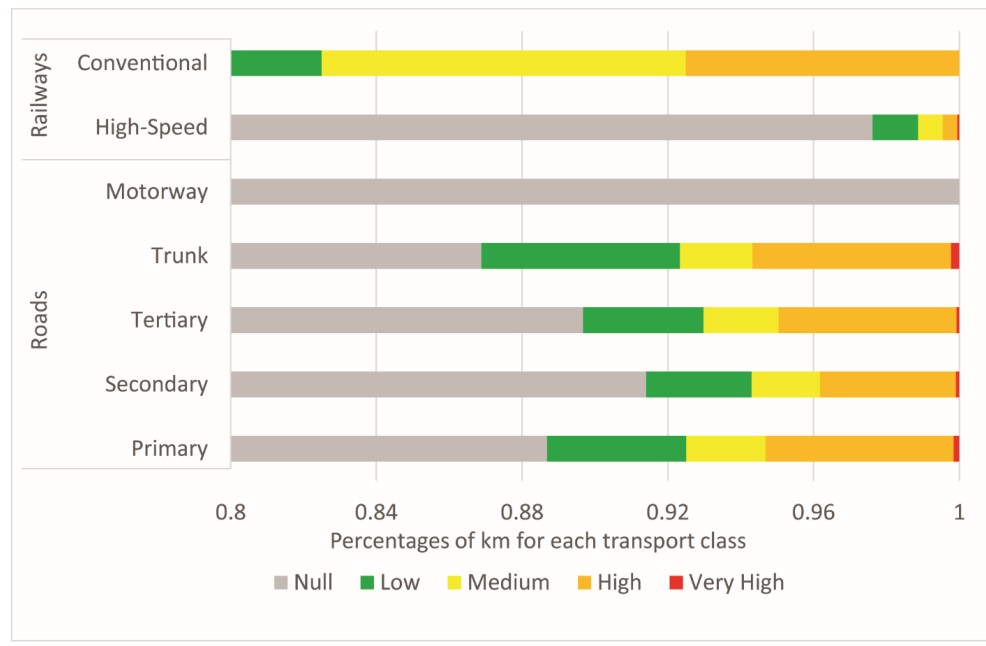

**Figure 17. Landslide susceptibility distribution for each transportation class.**

## 5. Discussion

### Landslide susceptibility

The main issue affecting the used random forest model is the need of an adequate training dataset to properly calibrate the predictor model. The first step of the work has been the homogenization of the landslide data, the used landslide inventory was created starting from different sources, hence, with quite non-homogeneous (e.g., in some cases the whole landslide perimeter was available, in other cases only a point representing the source area




of each landslide was provided, without info about the landslide dimension or propagation distance; more in
general there were few or no data about the landslide type or triggering causes). The lack of some data about the
landslides, or the partial or complete lack of landslides as in Kazakhstan and Turkmenistan, could lead to
underestimate the real landslide hazard of the studied countries, since some points could have been wrongly
classified (e.g., they have been considered as no landslide areas, but it was possible that a not reported landslide
was present). Furthermore, not all the adopted landslide inventories included information regarding the landslide
types, leading to the creation of a general landslide susceptibility map, where all the types of landslides are
considered. The created maps have been validated only using the available landslide dataset, providing good results
and highlighting the good prediction capability of the model. Anyway, an in-situ validation in some sample areas
can help to verify the quality of the results. As previously stated, for Turkmenistan there was no landslide inventory
available to train the RF model, therefore the corresponding LSM was obtained applying the model trained for the
other four countries. The lack of landslide data did not allow any validation of the result or estimation of the quality
of the susceptibility map of Turkmenistan. Furthermore, applying the model developed for the other countries, the
same importance of the conditioning factors (e.g., the independent variables) was assumed. For these reasons, the
landslide susceptibility map for Turkmenistan is more uncertain than those evaluated for the other four countries.
Among the used conditioning factors, soil type, distance from roads and distance from hypocentres resulted to be
the most influencing factors in slope stability, while planar curvature resulted with a high variability of its
importance. These parameters have been hence more deeply analysed to understand how the influence landslide
susceptibility. According to the partial dependency plots (Fig. 18), which show how the values of each
conditioning factor influence the landslide susceptibility, the soil types more related to landslides are lithosols and
cambisols, low thickness soils limited in depth by a continuous coherent and hard rock layer, located in steeply
slopes, with more than 30% of slope gradient. While the classes that have the lowest importance score are fluvisols
(young soils in alluvial deposits), xerosols (mainly arid clay) and chernozems (soils rich in organic matter), each
situated in flat to hilly areas, with less than 30% of slope gradient. Distance from roads, as expected, is important
for low values since the importance score is maximum for distance close to zero and it decrease exponentially with
the increasing of the distance. A similar behaviour can be noted with the distance from hypocentres, meaning that
areas close to hypocentres (within a radius of about 25 km) can more easily experience landslide phenomena in
case of future earthquakes. The partial dependency plot of planar curvature showed that the variability highlighted
in Fig. 16, is in fact, not so relevant since the range of the importance score is quite limited (values ranging from
0.4992 to 0.5008). In addition, it is possible noticing that negative values of planar curvature have a higher
importance score than zero values or positive values, meaning that concave slopes are more prone to landslide
than plain or convex surfaces.
***Landslide susceptibility and exposed elements***
The integration of susceptibility map with the maps of element al risk and communication router allowed to
identify those elements potentially more prone to landslide hazard, even of with some limitations. The obtained
results are indeed influenced by the input data (the susceptibility maps and the elements at risk databases).




The buffering procedures on roads and railways could overestimated or underestimated the susceptibility
distribution in some cases, likewise the analysis at 1km$^2$ resolution on population and buildings could led to an
exaggeration in the assessment of elements distribution in each class of landslide susceptibility.

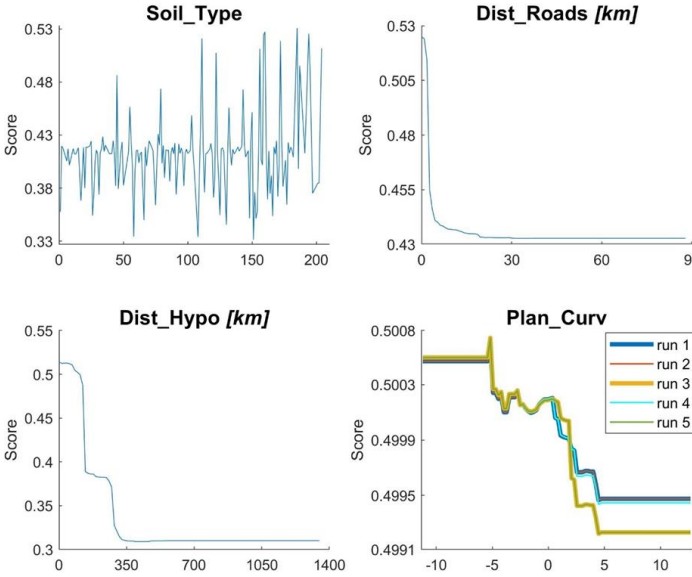


**Figure 18. Partial dependence plots.**

Nevertheless, the adopted approaches represented the only way to obtain an analysis as much accurate as possible
respect to the input databases. In this perspective, the detail of analyses could be improve focusing both on the
refinement of the analysis resolution (e.g., population and buildings) and on the elements at risk that are not located
in flat areas, where the landslide susceptibility is surely 0 or NULL.
**6. Conclusions**
In this work a new landslide susceptibility assessment of Central Asia was carried out. With respect to previous
works, in this research a unique map was created and with a higher resolution, in order to avoid boundary effects,
to get to more homogeneous and with better resolution results. The used approach allowed also to identify the
most relevant landslides predisposing factors: soil type distance from roads and hypocentres. The size and the
heterogeneity of the study area required the use of many input variables, some of them never used before in
landslide hazard assessment, and the elaboration of a high volume of data, as well as the adoption of some specific
procedure to accounting for the presence of heterogeneities and uncertainties in the input data, as the presence of
point landslides. The comparison with elements at risk and communication routes allowed a better assessment of
landslide hazard in the area, which can be useful to improve the land use management and to reduce the risk. The
main limitation of the work is related to absence of data about type and geometry for several landslides; in the
future a better input landslide inventory could help get to different susceptibility maps for different landslide types.
Another limitation is due to the absence of any information about the presence or absence in Turkmenistan, which
did not allow any clear validation of the results for this country.
*Code and data availability*. The landslide susceptibility model was implemented by using the cited landslide
inventory maps, published by the following authors: Behling et al., 2014, 2016, 2020; Havenith et al., 2015a;
Kirshbaum et al., 2015; Pittore et al., 2018; Strom and Abdrakhmatov, 2018. Other data implemented in the model,
such as MERIT DEM, geological formations, Active Fault Database, soil type map, rainfall maps are available
from Yamazaki et al. 2017, Persits et al., 1997, Styron and Pagani, 2020, https://land.copernicus.eu/,
www.ecmwf.int/en/forecasts/datasets/reanalysis-datasets/era5, respectively. The database on infrastructures, river
network, PGA and other landslide inventories were provided by the SFRAAR project partners: RED (Risk,
Engineering + Development – Pavia, Italy), OGS (National Institute of Oceanography and Experimental
Geophysics, Seismological Research Center, Trieste, Italy), IWPHE (Institute of Water problems, Hydropower,
Engineering and Ecology, Dushanbe, Republic of Tajikistan), ISASUZ (Institute of Seismology of the Academy
of Science of Uzbekistan, Tashkent, Uzbekistan), LLP (Institute of Seismology of the Science Committee of the
Republic of Kazakhstan, Almaty).
*Author contribution*. Ascanio Rosi implemented the landslide susceptibility model, William Frodella conceived
with Ascanio Rosi the article structure and collected the data, Nicola Nocentini prepared the landslide
susceptibility data and supported the model implementation, Francesco Caleca prepared the infrastructure data for
the model implementation and performed the statistical analysis involving the landslide susceptibility areas and
the exposed elements. All the above mentioned authors contributed to the writing of the article and the figure
graphics. Hans Balder Havenith and Alexander Strom provided the landslide databases from: i) Havenith, H.B.,
Strom, A., Torgoev, I., Torgoev, A., Lamair, L., Ischuk, A., Abdrakhmatov, K.: Tien Shan geohazards database:
Earthquakes and landslides. Geomorphology 249, 16–31, 2015a; and ii) Strom, A., Abdrakhmatov. K.: Rockslides
and rock avalanches of Central Asia: distribution, morphology, and internal structure. Elsevier, 441pg. ISBN: 978-
0-12-803204-6, 2018. They also provided the landslide pictures for figures 3 and 4, and critically reviewed the
paper. Veronica Tofani coordinated the work and reviewed the paper.
*Competing interests*. The contact author has declared that none of the authors has any competing interests.
*Acknowledgements*. This work was developed within World Bank-funded project "*Strengthening Financial*
*Resilience and Accelerating Risk Reduction in Central Asia*" (SFRARR), in collaboration with the European Union,
and the GFDRR (Global Facility for Disaster Reduction and Recovery), with the goal of improving financial
resilience and risk-informed investment planning in the central Asian countries (Kazakhstan, Kyrgyz Republic,
Tajikistan, Turkmenistan and Uzbekistan). This work represents the outcomes of the Task 7 "Landslide Scenario
Assessment", coordinated by the UNESCO Chair on Prevention and Sustainable Management of Geo-
Hydrological Hazards (University of Florence). In particular, the authors would like to thank Gabriele Coccia and
Paola Ceresa from Red Risk Engineering (Pavia, Italy) for providing river network data and for the valuable
coordination and constant support, as well as Valerio Poggi and Chiara Scaini from OGS (National Institute of
Oceanography and Experimental Geophysics, Seismological Research Center, Trieste, Italy) for providing seismic
data (active faults, Pga) and exposure data. We would also like to thank the partners from Central Asia for the
fruitful collaboration, in particular: IWPHE (Tajikistan), ISASUZ (Uzbekistan) and LLP (Kazakhstan,).

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
