# Peer review of "COMPREHENSIVE LANDSLIDE SUSCEPTIBILITY MAP OF CENTRAL ASIA"

_Natural Hazards and Earth System Sciences, 2023_

## Referee Comment (RC1)

Review of the manuscript '***COMPREHENSIVE LANDSLIDE SUSCEPTIBILITY MAP OF CENTRAL ASIA***' by *Rosi et al.* submitted to *Natural Hazards and Earth System Sciences*.

Recommendation: ACCEPT

Focus of the paper: landslide susceptibility mapping aimed at land use-planning and risk reduction strategies in the Central Asia region which is prone to landslides due to high tectonic activities.

Relevance: The presented study is the original primary research within scope of the journal. The manuscript meets general criteria of the significance in Earth science and geologic risk assessment The study has been conducted in accordance to the technical standards in spatial analysis, mapping and numerical assessment. It is relevant to the journal topic as corresponding to the major domain and research disciplines: Geography, Earth sciences, Hazard risk assessment, and Landslide mapping.

Abstract is well written and clearly describes the undertaken study. It describes the major problems of Central Asian region related to tectonics and seismicity which results in high risk of landslides. Methods used in this study are mentioned, results reported briefly and concisely. Abstract is well done.

Structure: The article is well organized with structured sections. The structure of the manuscript conforms to the journal standards and discipline norm. It has the following standard sections: *1. Introduction; 2. Study area; 2.1 Landslide types in Central Asia; 2.2 Large Rockslides and natural dams; 2.3. Landslide in soft rocks and loose deposits; 3 Materials and Methods; 3.1 Landslide databases; 3.2 Random Forest (RF) model; 3.3. Selection of independent variables; 3.4. Model optimization; 3.5. Model training; 3.6. Model validation; 3.7. Landslide susceptibility and elements at risk; 4. Results; 4.1 Susceptibility map; 4.2 The Fergana valley mountainous rim; 4.3 Trained model performances and conditioning factors relevance; 5. Discussion; 6. Conclusions*. Some sections are divided into the minor subsections and paragraphs for a better structure. The numeration of the sections is correct and consecutive.

Logic: The clarity of the text logic and organization of the paper is sufficient. It demonstrates the consistent interpretation of the results with detailed explanations and comments. A comparison of the results with those in previous studies regarding landslide risk assessment is presented.

Introduction presents a background, defines research goals and provides a clear statement of research problem on landslide disasters in Central Asia. *Rosi et al.* reported economic losses and damages caused by landslides and presented the observed landslide hazard statistics summarised in a table. It is clear from this table that the most affected region is Tajikistan. The authors then compared the effects from landslide and discussed briefly the outcomes for other countries in Central Asian region. Existing studies on regional studies on landslide hazard are mentioned as well as initiatives on landslide inventories. *Rosi et al.* then overviewed the existing methods of landslide risk assessment and approaches to susceptibility mapping, among which methods of machine learning. They then introduced the approach use din their study – Random Forest algorithm, and mentioned its advantages. The Introduction well describes the research. Introduction and background show context of the article. Literature is well referenced and relevant.

Study area: is described with sufficient details in section 2. Maps depicting the target region are inserted. *Rosi et al.* described landslide types in Central Asia with sufficient details in subsection 2.1. Rockslides existing in the region with Palaeozoic magmatic and metamorphic crystalline bedrock are described with examples presented in photos. Study area section is well done.

Research questions and goals are well identified: presenting maps of landslide susceptibility in

Central Asian region using methods of machine learning. Objectives are relevant and meaningful with regard to natural risk assessment in the geologically unstable region prone to earthquakes and landslides. It is stated how the research fills an identified knowledge gap by presenting new advanced methods for modelling landslide risks.

Literature regarding the relevant topics is reviewed, formatted according to the journal rules and appropriately referenced. Major sources include published papers on natural risk assessment, with a special focus on landslide and avalanches, analysis of risks in seismically active regions; papers on applied statistics and issues of disaster risk management and climate-related issues. The authors also reviewed papers on regional studies with regard to Central Asian region.

Research gaps and weakness in former works are described; the existing gaps are identified. The contribution of this work filling this gap is explained. It concerns providing novel methods applied to landslide risk evaluation and modelling.

Motivation is explained: this study contributes to fill in the gaps in the existing similar research through presenting a unique accurate map created with a higher resolution compared to existing works, aimed at landslide susceptibility assessment of Central Asia. Given the size and heterogeneity of the study area, *Rosi et al.* used many input variables including landslide inventory which ensured the accuracy of modelling. the authors also considered the elements at risk and communication routes when assessing the risks.

English language: acceptable. Clear, unambiguous, professional English language used throughout.

Data used in this study are described and summarise with a comprehensive details in subsection 2.1. Landslide databases (pp. 8-9). The research presents novel data and results regarding landslide inventory and data derived from reliable repositories. Data are explained, sources are mentioned. The key parameters used for modelling are listed in subsection 3.3. Selection of independent variables (pp. 10-11).

Methods: Methods described with sufficient detail and information to replicate. *Rosi et al.* identified many key parameters of landslide susceptibility such as soil type, lithology, elevation, the distance from roads, rainfalls and slope curvature, among others. This supports the accuracy of the assessment and modelling. Modifications of the existing methods are mentioned briefly. The workflow is well structured and clearly described with sufficient information to reproduce the approach. The authors used advanced methods of modelling using machine learning algorithms of Random Forest (RF), and LSM. The models were defined and optimised, trained, executed and validated (true/false).

Results are reported: *Rosi et al.* assessed the data by modelling approach and presented the results. The results include the obtained Landslide susceptibility map of Central Asia (Figure 9); Detail of the landslide susceptibility map with the overlapping landslide polygons (Figure 10); Landslide susceptibility class intervals summarised in Table 2; Details of the landslide susceptibility map (Figure 11); Landslide susceptibility map obtained for Fergana (Figure 12) and Frequency histogram of susceptibility classes (Figure 13). The results are well presented, convincing and argued. The Results are presented with clarity and include description, graphical illustrations, tables, and descriptions. The results are relevant to the initial research goals and objectives and highlights major achievements of this study.

Discussion interpreted the major outcomes of this study. The advantages of the obtained results are described and compared with other studies. The Discussion described the issues of methodology and results.

Conclusion Conclusions are well stated, linked to original research question, limited to supporting results and summarized the study with interpretation of facts. The conclusions are appropriately stated and connected to the original questions.

Actuality, novelty and importance of the research is clear. It consists in technical approach of landslide susceptibility and risk assessment and evaluating the over the Central Asian region using multi-source geospatial datasets and advanced methods of modelling (RF model and LSM model).

Academic contribution: Rigorous investigation performed to a high technical and professional standard. The paper increases the knowledge in landslide risk assessment in Central Asian region. The paper combines technical approaches, cartographic work and machine learning algorithms which presents a multi-disciplinary study well deserved to be published in *NHESS*.

Figures The authors presented 18 figures which are of acceptable quality, easy to read, relevant and suitable. Figures are labelled and appropriately described. They illustrate the results of the undertaken study.

Recommendation: This manuscript can be ==ACCEPTED== based on the detailed report above.

With kind regards,

- Polina Lemenkova.

15.03.2023.

---

## Author Response (AR1)

This is an interesting paper on landslide susceptibility mapping in Central Asia. I really enjoyed reading this paper, given the challenge of mapping landslide susceptibility over such a large area, and was interested in how the authors dealt with their very complex (heterogeneous) and large data sets. Also, the random forest method seems to be well suited for such an approach on a very large scale. The paper is proposed by an internationally recognized research team with strong experience in landslides. I found it well written and easy to understand. It is an original paper within the scope of the journal that brings novelty to a regional case study. Even if the methods and approach are not specifically new, the application on such a large scale is challenging and worth mentioning.

I recommend it to be accepted with some revisions.

Dear reviewer, thank you very much for your constructive comments and useful suggestions, we modified the paper accordingly. We believe that the quality of our paper has now improved.

1. Major comments

I am not really convinced by the part that consists in analyzing the spatial correspondence between the landslide susceptibility and the elements at risk. To me this part is not really "in the line" with the rest of the article which already brings interesting (and sufficient) technical elements on the process used to map landslide susceptibility all results and discussions that follows. I don't fundamentally disagree with the approach, and several authors in quite recent research projects showed that at large scale, the risk analysis can not clearly include the temporal dimension (when and how frequently). Then, risk maps are often only the combination of susceptibility and elements at risk. See for example the handbooks of the CHARIM project by van Westen et al. https://www.cdema.org/virtuallibrary/index.php/charim-hbook/methodology/4-landslide-hazards/4-3-landslide-susceptibility-at-the-national-scale

Nevertheless, I think the message of the article would be clearer without this risk part (or maybe just evoked as a prospect or ongoing work in the discussion). Otherwise, the title of the paper should be changed to "Comprehensive landslide susceptibility map and risk assessment of central Asia" (or something similar) and the description of the results should be developed a bit more. This is the reason why I considered a major revision for this paper.

We discussed this issue with the other authors and we decide to remove the section regarding the exposed elements

2. Specific Comments

2.1. Landslide data harmonization

The part dealing with landslide inventory harmonization is interesting (end of section 3.4, lines 343 to 356). To me, it may require a specific subsection that could be called "Landslide Inventory Harmonization" (or some similar idea) to focus specifically on this point, which to me is critical in this research. Perhaps an additional figure (methodological sketch) on this specific point (i.e. buffer zones/omitted training no landslides vs. polygon mapped landslides etc.) will help to highlight this part of the process.

Also in this section, I couldn't find any information on how the polygon mapped landslides were integrated into the RF training. Are these landslides mapped with separate polygons for triggering and runout areas? Traditionally, landslide susceptibility maps are made based on the triggering area only, with the runout or accumulation zone generally considered to be affected, but not strictly "causing" the landslide. At this scale

it may be less important, but still, can you please add this information and discuss it if you feel it is appropriate.

Apparently, the authors used all possible "points" or pixels falling into a landslide/triggering area (or used a buffer approach for point features) to train the RF model. Using statistically based methods (which I personally know better that RF), we generally use only one "point" per landslide considering each landslide as an event. This consideration is also recommended to avoid to artificially increase the weight or influence of large landslides in the calculation. However, I don't know how sensitive RF is to this issue.

Dear reviewer, thanks for your observations. Since the landslide inventory was made both of polygons and points, it was not possible to divide each landslide into source, track, and deposition zone. The exact position of points on landslides was not known as well, we have been referred by the project partners that points represent the centre of the landslide area, but without any data about the approximation level. Because of the paucity of landslide data (with respect to the extension of the study area), we considered all the pixels inside the polygons and a 100 m buffer around the landslide points (please refer to section 3.5 row 374 "all the points within a landslide"). As we wrote in the manuscript, we removed overlapping landslides, to avoid over estimation and weighting of certain areas.

We selected RF model because it is not very susceptible to overfitting issue, since it considers each pixel independently (the relative position and distance between pixels is not considered, since we did not give any info to the model about this), so the dimension of the landslides does not apply any weight to the variables.

We modified the structure of the manuscript adding sections 3.4.1, 3.4.2 and 3.4.3 and we added a flow diagram of the procedure.

2.2. Explanatory factors

In general, rainfall data are not used for susceptibility analysis, since it is more a matter of triggering and thus hazard. However, several large papers have used it anyway, as it could be considered a preparatory factor under some conditions (and it is actually mentioned in your landslide description section). I think you could add a sentence or two to justify more precisely why and how you decided to include it (e.g. around line 300).

Thanks for the suggestions. We added more text in section 3.3.

2.3. Results section

Could you please justify the close-up view of the selected study area? I can easily understand this, but a transition sentence would be welcome.

The close-ups have been used to better show the result of the analyses, since it is almost impossible to appreciate the details with a global view of the study area. We added a sentence after table 2. Fergana valley close-up was added since it is a very relevant area, according to the local partners, which is worth of a dedicated sub-section.

2.4. Landslide data presentation

Even though the different landslide inventories are well presented in the different paragraphs of section 3.1, I think a summary table mentioning references, area, number of mapped landslides, scale, data format (i.e. point, polygon or other) etc. would be a nice summary for the readers.

We have added a new Table 2.

2.5. ROC curves

I didn't understand if the ROC curves in Figure 14 were done for training samples or validation or both? In general, we show calibration ROC curves (from training samples) and validation ROC curves (from independent sample). Could you please clarify this?

The ROC curves reported in fig. 14 represent the validation dataset; we modified the caption of the figure for more clarity

3. Technical corrections

Thanks for the comments, we modified the text and the figures accordingly. Fig. 11 was also improved, while the conclusions were rewritten. Minor errors were corrected thorough tough the manuscript.

Line 158 : a dot is missing at the end of the sentence

Line 422 : capital letter missing

Line 564 : "they" instead of "the" (I guess)

Line 580 : "at" Instead of "al"

Line 580 : "router?" I'm not sure this is right. "Route?" "Road?"

Fig 5 title  "map of the adopted landslide inventory"

Fig 5. This figure is zoomed to a specific region that is difficult to locate within the entire study area. You may add the countries names to help reading it.